# Structural basis for Gemin5 decamer-mediated mRNA binding

Qiong Guo[1,6], Shidong Zhao[1,6], Rosario Francisco-Velilla [2,6], Jiahai Zhang[1], Azman Embarc-Buh [2], Salvador Abellan [2], Mengqi Lv[1], Peiping Tang[1], Qingguo Gong[1], Huaizong Shen [3], Linfeng Sun [1], Xuebiao Yao [1], Jinrong Min [4,5], Yunyu Shi[1], Encarnacion Martínez-Salas [2] ✉, Kaiming Zhang [1] ✉ & Chao Xu [1] ✉

Gemin5 in the Survival Motor Neuron (SMN) complex serves as the RNA-binding protein to deliver small nuclear RNAs (snRNAs) to the small nuclear ribonucleoprotein Sm complex via its N-terminal WD40 domain. Additionally, the C-terminal region plays an important role in regulating RNA translation by directly binding to viral RNAs and cellular mRNAs. Here, we present the three-dimensional structure of the Gemin5 C-terminal region, which adopts a homodecamer architecture comprised of a dimer of pentamers. By structural analysis, mutagenesis, and RNA-binding assays, we find that the intact penta-mer/decamer is critical for the Gemin5 C-terminal region to bind cognate RNA ligands and to regulate mRNA translation. The Gemin5 high-order architecture is assembled via pentamerization, allowing binding to RNA ligands in a coordinated manner. We propose a model depicting the regulatory role of Gemin5 in selective RNA binding and translation. Therefore, our work provides insights into the SMN complex-independent function of Gemin5.

After completing gene transcription, eukaryotic mature mRNAs are exported from the nucleus to the cytoplasm[1], where mRNAs associate with various RNA binding proteins (RBPs) that play important roles in the stabilization, localization, and translation of mRNAs[2,3]. Work done over the years has provided strong evidence for the role of specific RBPs in the regulation of gene expression at the posttranscriptional level via association with the translation machinery[4,5].

The cytoplasmic survival motor neuron (SMN) complex associates with small nuclear RNAs (snRNAs) and facilitates their assembly into small nuclear ribonucleoproteins (snRNPs)[6,7]. Gemin5, the RBP in the SMN complex, is a 170 kDa protein of 1508 amino acids containing an N-terminal WD40 domain and a C-terminal α-helix rich region[8–10]. It has been recently shown that biallelic mutations in the *Gemin5* gene, most of which are located in the C-terminal region, lead to human

neurodevelopmental disorders[11–13]. These patients, however, show phenotypic features apart from those originating from defects in the SMN protein causing spinal muscular atrophy (SMA).

We and other groups have reported that the Gemin5 N-terminal WD40 domain spanning residues 1–739 specifically recognizes both the m⁷G cap and the Sm site within snRNAs[10,14–17]. In addition to its SMN-dependent role in snRNA delivery, Gemin5 also possesses functions outside of SMN complexes. It has been reported that during foot-and-mouth disease virus (FMDV) infection, Gemin5 undergoes proteolysis to generate a transient C-terminal fragment spanning residues 845–1508[18], termed G5C herein. Thus, following cleavage, G5C displays biological functions independent of the Gemin5 N-terminal WD40 domain (G5N). Early work showed that the full-length Gemin5 protein interacts with internal ribosome entry site (IRES) elements of two

[1]MOE Key Laboratory for Cellular Dynamics, School of Life Sciences, Division of Life Sciences and Medicine, University of Science and Technology of China, 230027 Hefei, China. [2]Centro de Biología Molecular Severo Ochoa, CSIC-UAM, Nicolás Cabrera 1, 28049 Madrid, Spain. [3]Key Laboratory of Structural Biology of Zhejiang Province, School of Life Sciences, Westlake University, 310024 Hangzhou, Zhejiang, China. [4]Structural Genomics Consortium, University of Toronto, Toronto, ON M5G 1L7, Canada. [5]Department of Physiology, University of Toronto, Toronto, ON M5S 1A8, Canada. [6]These authors contributed equally: Qiong Guo, Shidong Zhao, Rosario Francisco-Velilla. ✉e-mail: emartinez@cbm.csic.es; kmzhang@ustc.edu.cn; xuchaor@ustc.edu.cn

viruses, FMDV and hepatitis C virus (HCV), to inhibit IRES-dependent RNA translation[19]. However, G5C performs different functions by binding to various types of RNAs[19–21]. Intriguingly, transient expression of an internal region of G5C encompassing residues 1287−1400, known as RBS1, associates with its own mRNA to increase translation efficiency[20,22], therefore counteracting the negative effect of Gemin5 on global translation. Thus, Gemin5 serves as a multifunctional RBP to achieve diverse functions through binding to different cognate RNA ligands via G5N and G5C[23].

Our previous work suggested G5C as a polymer[14], and more recently, the crystal structure of the TPR domain, spanning residues 845−1096 within G5C, was found to form a homodimer[24]. However, given that the TPR domain alone does not display detectable RNA binding capacity and that the identified RBS1 comprises a non-conventional RNA binding module[24,25], the molecular mechanism underlying the G5C−RNA interaction remains elusive.

To gain structural insights into the SMN-independent functions of Gemin5, we determined the near-atomic resolution structure of G5C by cryogenicelectron microscopy (cryo-EM). We find that the C-terminus of G5C forms a pentamer, which further dimerizes via the TPR module to adopt a homodecamer (a dimer of pentamers). The protomers within the pentamer establish extensive hydrophobic interactions with each other, which are validated by biochemistry and mutagenesis experiments. Furthermore, by using in vitro RNA binding experiments and in vivo RNA translation, we show that an intact G5C decamer is required for binding to its cognate RNA ligands, and destabilization of the pentamer/decamer impairs both RNA binding and mRNA translation. Therefore, our work sheds light on understanding the role of the G5C decamer as a mediator in mRNA translation outside the SMN complex.

## Results

### Self-assembled G5C polymer binds to different RNA fragments

To study the architecture and RNA binding properties of G5C, we cloned, expressed, and purified G5C spanning residues 841–1508 (Fig. 1a). Recombinant G5C was homogeneous after purification by anion-exchange column, and consistent with our previous work[14], it is eluted as a high molecular weight (MW) polymer in size-exclusion chromatography (SEC). Static light scattering (SLS) experiments indicated that the molecular weight (MW) of G5C in solution is ~740 kDa with a polydispersity of 1.007 (Supplementary Fig. 1).

It has been reported that the RBS1 domain of G5C (residues 1287–1400) recognizes different RNAs, including domain 5 (D5) of the FMDV IRES element and the stem-loop 1 (SL1) region of Gemin5 mRNA (NM_015465.5) spanning nt 3959–4044[20,21,25]. By using an in vitro transcription assay, we synthesized SL1, an RNA fragment derived from Gemin5 mRNA (Fig. 1b). An electrophoretic mobility shift assay (EMSA) showed that G5C binds to SL1, resulting in two retarded bands (Fig. 1c). We also synthesized a short RNA fragment (D5) derived from the FMDV IRES and found that it bound G5C with weaker affinity than SL1 (Figs. 1d–f).

Next, we used a fluorescence polarization (FP) binding assay to quantitatively examine the $K_D$s between G5C and different RNAs. The binding data indicated that G5C bound fluorescein amidite (FAM)-labeled SL1 with a $K_D$ of 6.5 μM, whereas a short form of SL1 (SL1[short]) spanning nt 3959–3990 bound to SL1 10-fold weaker than SL1 ($K_D$s: 67 μM vs. 6.5 μM) (Fig. 1g). As a negative control, G5C does not display detectable binding toward FAM-labeled poly U single-stranded RNA (U₇) ($K_D > 500$ μM) (Fig. 1g). Collectively, our data suggest that G5C assembles into a high molecular weight complex in solution, which binds to previously identified RNA ligands.

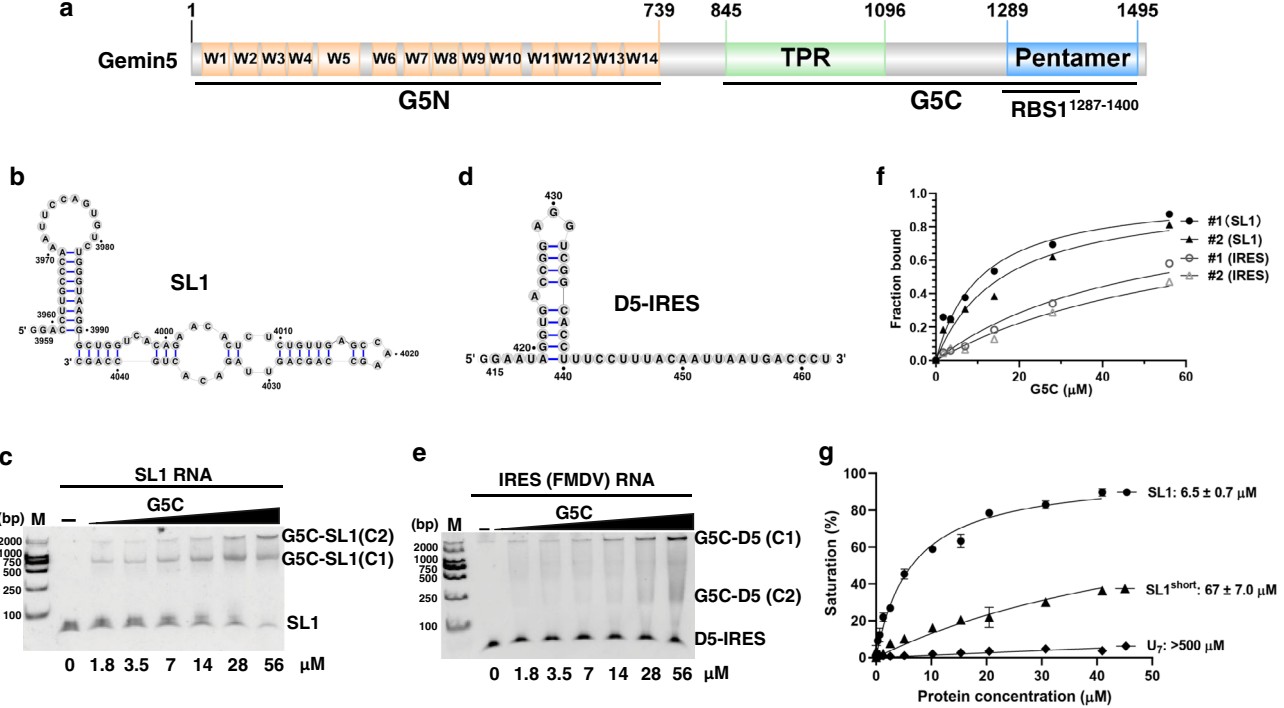

**Fig. 1 | Purified G5C (Gemin5₈₄₁₋₁₅₀₈) binds to SL1 mRNA and the IRES region of FMDV viral RNA. a** Domain architecture of human Gemin5, with the WD40 repeats, TPR domain, and C-terminal pentamer region shown in orange, green, and blue, respectively. The RBS1 region within G5C (aa 1287–1400) is annotated. **b** The sequence and predicted structure of the SL1 region derived from Gemin5 mRNA. **c** EMSA binding assay between purified G5C and SL1 RNA. The experiment was repeated independently twice with similar results. **d** The sequence and predicted structure of the IRES D5 region (D5-IRES) derived from FMDV viral RNA. **e** EMSA binding assay between purified G5C and D5-IRES. The experiment was repeated independently twice with similar results. **f** Binding curves of duplicate EMSA for G5C binding to SL1 RNA and D5-IRES, with the X- and Y-axes indicating the G5C concentration and fraction bound, respectively. **g** FP binding assay for G5C binding to SL1, SL1[short], and poly-U (U₇) RNAs with the indicated $K_D$ values. Data represented as mean ± SD (n = 2 per concentration and two individual experiments). In Fig. 1c and 1e, C1 and C2 represent lower and upper band, respectively.

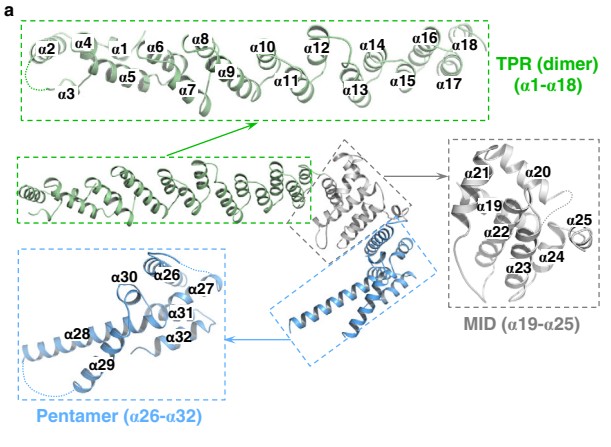

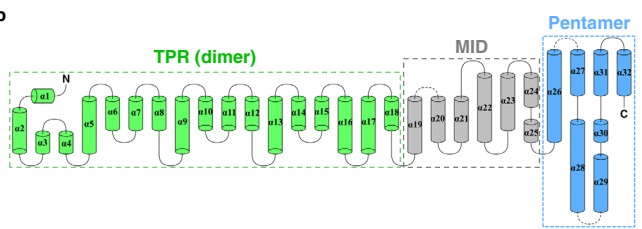

**Fig. 2 | Three-dimensional structure of the G5C protomer. a** The TPR (α1–α18), MID region (α19–α25), and pentamer region (α26–α32) are shown in green, gray, and blue, respectively. The invisible loop regions are indicated with dashes. **b** Topology diagram of G5C with secondary structures is colored in the same way as in (**a**).

## G5C assembles into a decamer comprised of a dimer of pentamers

To understand the mechanism underlying G5C self-assembly, we solved the structure of G5C by cryo-EM at an overall resolution of 3.31 Å based on the gold-standard Fourier shell correlation (FSC) curve. Local resolution analysis demonstrated that the resolution of visible G5C regions was within 3.6 Å (Supplementary Figs. 2–4, Supplementary Table 1). Symmetry expansion and focused refinement for the G5C protomer generated an overall map of 2.6 Å resolution. Given the high quality of the cryo-EM map, we performed de novo model building for the other G5C regions after the dimerized TPR domains (PDB: 6RNQ)[24] were docked into the map. Consistent with the result from the SLS experiment, G5C forms a homodecamer with an MW of ~750 kDa (Supplementary Fig. 1). For each protomer, 531 out of 663 residues were unambiguously built (Fig. 2a), with several unresolved loop regions in the structure due to intrinsic flexibility.

The G5C protomer is composed of 32 α-helices arranged in three regions (Fig. 2a, b). The N-terminal region of G5C consists of α1–α18, with α18 packing against α17 of the solved TPR domain (α1–α17) to form an extra TPR module (Fig. 2a). The middle region is composed of α19–α25. α22–α24 constitute a helix bundle, with α22 and α24 packing against α19–α21. The loop between α19 and α20 (aa 1133–1154) is invisible in the structure. α25 extends from α24 and connects the middle region to the C-terminal region (Fig. 2a). The C-terminal region consists of α26–α32 containing two helix bundles, α28–α29 and α26–α27–α31–α32. α30 is a short helix packing with α26, α28, and α29. The loops between α26 and α27 (aa 1294–1345) and between α28 and α29 (aa 1392–1429), as well as the last 12 C-terminal residues (aa 1497–1508), are invisible in the structure (Fig. 2a).

To our surprise, five G5C molecules (A–E or A′–E′) form a pentagon-like structure via its C-terminal region, named the pentamer region thereafter (Figs. 1a, 2, and 3a). Consistent with the previously solved TPR dimer structure[24], A–E forms homodimers with A′–E′. Thus, 10 G5C molecules assemble into a decamer

comprised of a dimer of pentamers (Fig. 3a). The side lengths of the outer and inner pentagons are 83 and 32 Å, respectively. The distance between the two parallel pentagon planes is ~177 Å with a rotation angle of 36° (Fig. 3a), and the rotation angle between the projections of two protomers in a TPR dimer, such as A and A′, is 108° (Fig. 3b). In summary, five G5C molecules are arranged into a pentamer with 5-fold symmetry, and the two pentamers further dimerize via TPR domains to form a decamer.

By searching the DALI server, we did not identify any homology structure that has >30% sequence identity to G5C. However, previously reported homodecamer structures, including those of NLRP3 (PDB id: 7PZC)[26] and cyanase (PDB id: 1DW9)[27], are characterized by a pentamer of dimers or a dimer of pentamers, which prompted us to compare them with the G5C decamer (Supplementary Fig. 5). For all decamer structures, the NLRP3 decamer adopts a pentamer of dimers (Supplementary Fig. 5a), whereas the G5C and cyanase decamers could be attributed to either a pentamer of dimers or a dimer of pentamers (Supplementary Fig. 5b, c). In all three structures, the modules that form the decamers are distinct. In addition, the rotation angles between the projections of the two pentagons are 108°, 36°, and 0° for Gemin5, NLRP3, and cyanase, respectively.

## Extensive hydrophobic interactions govern pentamer formation

At the protomer–protomer interface of the G5C pentamer, an intermolecular five-helix bundle is formed by α28 and α29 of one protomer (molecule A, red) and α26, α31, and α32 of another protomer (molecule B, cyan), mainly via hydrophobic interactions (Fig. 3c), with a total buried accessible surface area of ~1400 Å². Specifically, Leu1381 of α28[A] makes hydrophobic interactions with Leu1465, Leu1468, and Leu1469 of α31[B]; Ala1382 and Met1384 of α28[A] contact Leu1469 and Leu1465 of α31[B], respectively; Ile1385 of α28[A] is snugly positioned into a hydrophobic pocket composed of Tyr1286 and Trp1289 of α26[B] and Leu1465, Val1466, and Leu1469 of α31[B]; His1388 of α28[A] makes van der Waals interactions with Leu1461 and Leu1465 of α31[B] and Leu1372 of α28[B]; Leu1431 and Thr1435 of α29[A] make additional hydrophobic interactions with Leu1465 and Leu1468 of α31[B] and Leu1490 of α32[B] to stabilize the complex (Fig. 3d). In addition to hydrophobic interactions, intermolecular hydrogen bonds or salt bridges are found between Gln1378[A] side chain amide and Ser1472[B] main chain carbonyl and between Gln1389[A] side chain amide and Glu1462[B] side chain carboxyl (Fig. 3d).

All residues involved in intermolecular hydrophobic interactions are conserved among eukaryotic species except the partially solvent-exposed Ala1382 of α28[A], suggesting that the pentamer architecture is conserved among Gemin5 orthologs (Supplementary Fig. 6). To validate the hydrophobic interface and to study whether pentamer formation is required for G5C binding to its RNA ligands, we made a double mutant (L1468D/L1469D) and a triple mutant (L1381D/M1384D/I1385D) by substituting conserved hydrophobic residues (Leu, Met, Ile) at the intermolecular hydrophobic interface (Fig. 3d) with a polar residue (Asp). Based on the SEC assay, neither the double mutant L1468D/L1469D (M1) nor the triple mutant L1381D/M1384D/I1385D (M2) assembles into an intact decamer, validating the hydrophobic pentamer interface (Fig. 4a, b). Since these mutants form a homodimer via the TPR domain and the mutations do not disrupt the hydrophobic interface completely, both mutants behave as a mixture of dimer and transient tetramer, as evidenced by the two elution peaks from the SEC assay (Fig. 4a, b). Interestingly, a single mutant, L1469H, altering a less conserved position of G5C, behaved as a decamer according to SEC (Fig. 4c). EMSA shows that both L1468D/L1469D and L1381D/M1384D/I1385D display much weaker SL1 binding affinities compared to G5C wild type (Fig. 4d, e), whereas L1469H retained RNA-binding capacity to SL1 RNA (Fig. 4f). These results indicated that assembly of the intact pentamer/decamer structure is required for G5C binding to its RNA ligands.

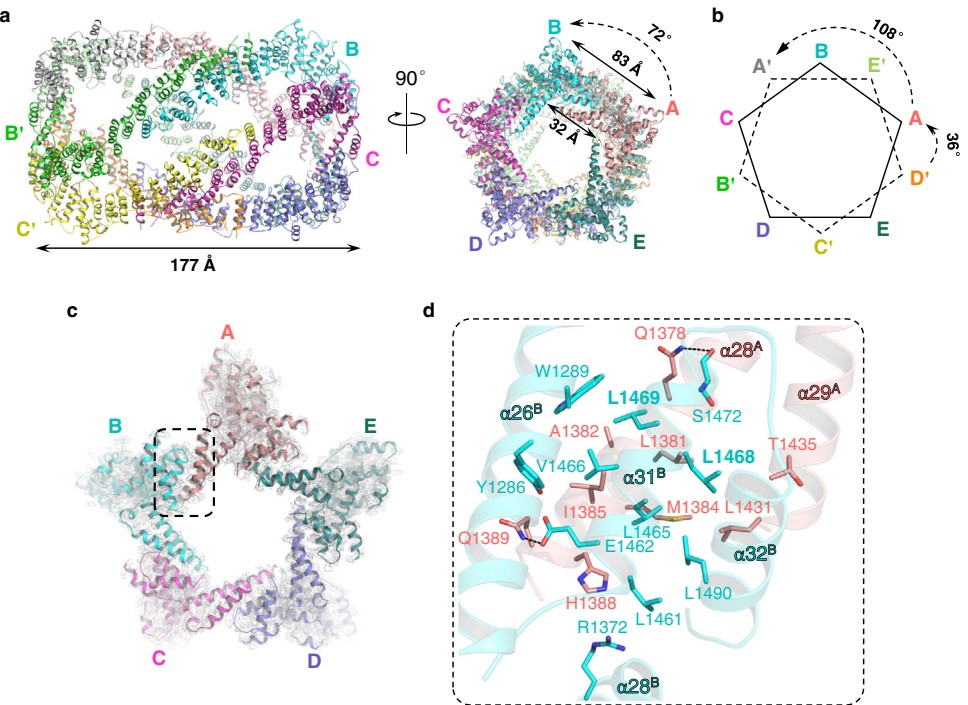

**Fig. 3 | Structure of the G5C decamer. a** Overall structure of the G5C decamer, with each molecule colored differently. The distance between the two parallel pentagon planes is -177 Å. The side lengths of the outer and inner pentagons are 83 and 32 Å, respectively. **b** Projection of two pentagon planes, with a rotation angle of 36°. **c** Structure of the G5C pentamer region, with the molecules colored as in (**a**).

The interface between molecules A and B is indicated by a dashed rectangle. **d** Close-up interaction of the protomer–protomer interface. Residues involved in intermolecular interactions are shown in sticks. Hydrogen bonds are shown in black dashes.

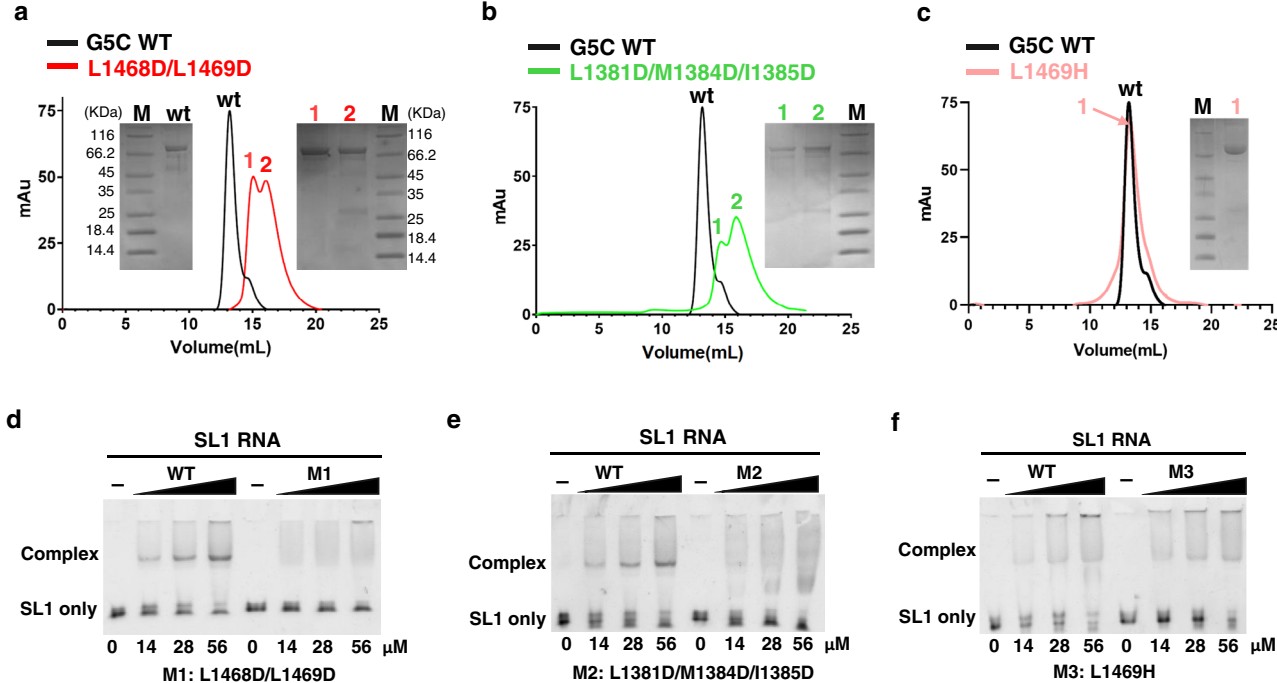

**Fig. 4 | Intact pentamer/decamer is required for G5C binding to SL1 RNA.**
**a**–**c** Size-exclusion chromatography (SEC) for G5C wild type and mutants, including (**a**) L1468D/L1469D (M1), **b** L1381D/M1384D/I1385D (M2), and **c** L1469H (M3). The same molecular weight marker was used in (**a**–**c**). **d**–**f** EMSA for SL1 RNA binding

with G5C variants, including (**d**) M1, (**e**) M2, and (**f**) M3. For each, wild-type G5C was used as the positive control. Each experiment was repeated independently twice with similar results.

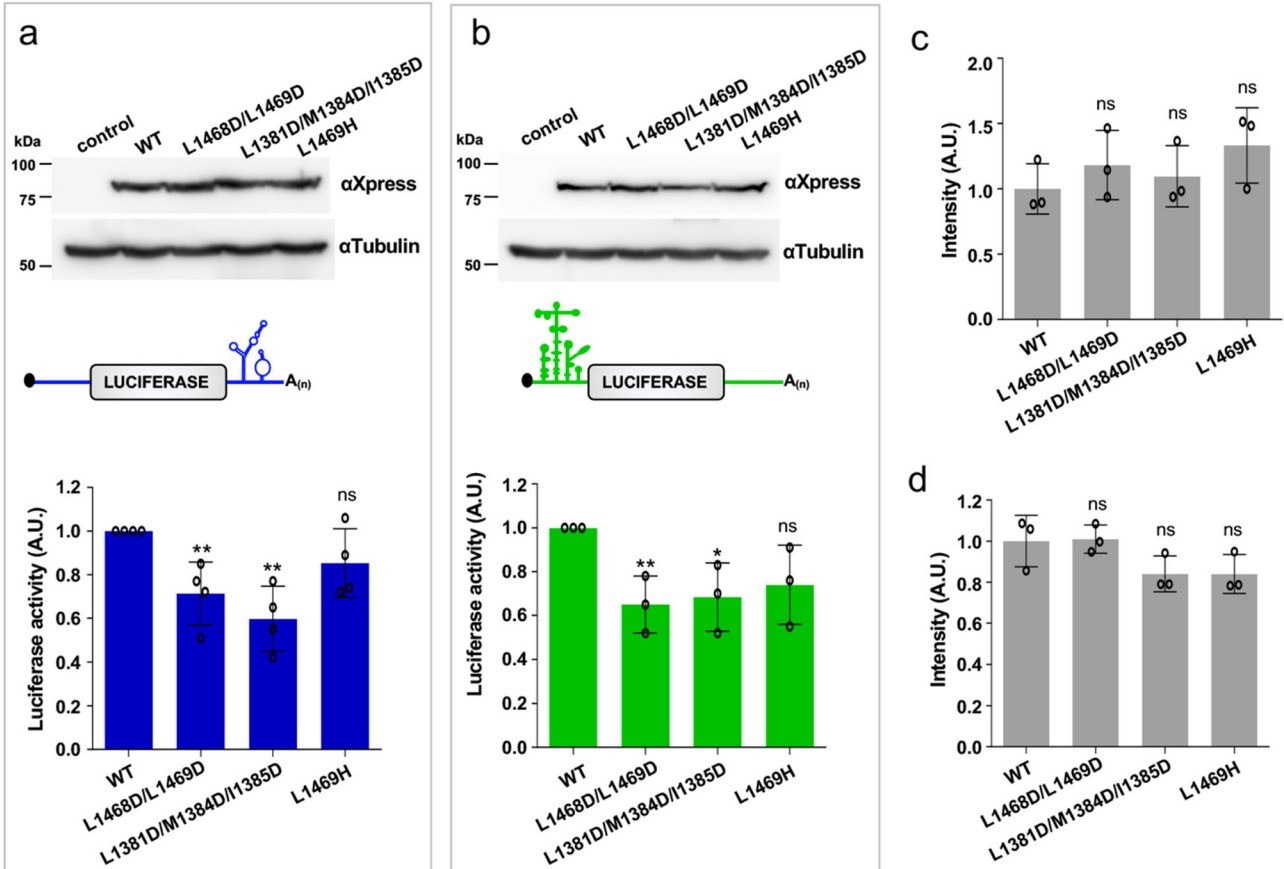

**Fig. 5 | Pentamer-destabilizing mutations impair G5C selective translation.**
Protein expression in HEK293 cells was monitored by WB using anti-Xpress for G5C proteins, tubulin was used as loading control. Diagrams of the luciferase reporter mRNAs used in the assay, **a** luc-SL1 and **b** IRES-luc. Luciferase activity was measured in cell lysates expressing the reporter mRNA, **a** luc-SL1 or **b** IRES-luc, co-transfected with Xpress-G5C-WT or the indicated mutants. Luciferase values are normalized to cells expressing Xpress-G5C-WT. Values represent the mean ± SD obtained in three independent assays using two-tailed paired Student $t$-tests. Comparison for the same: **$p < 0.01$, *$p < 0.05$; ns, not significant. Specific $p$ values are 0.007124, 0.001610, 0.104045 in the indicated order for (**a**) luc-SL1 translation, and 0.009374, 0.022961, 0.065317 for (**b**) IRES-luc translation. **c**, **d** Values represent the mean ± SD obtained in three independent assays using two-tailed paired Student $t$-tests. Comparison is for the same: ns, not significant. The specific $p$ values are 0.389685, 0.611948, 0.171729 for (**c**), and 0.739000, 0.059173, 0.955913 for (**d**).

## Pentamer-destabilizing mutations impair Gemin5-mediated translation

Gemin5 plays an important role in selective mRNA translation by binding to specific stem-loops of its own mRNA, as well as to other mRNAs[25,28,29]. To study whether the pentamer-destabilizing mutations impairing RNA binding also have an impact on in vivo translation, we examined their roles in translation in HEK293-transfected cells (ATCC, CRL-1573) using two different reporters. One harbors the SL1 motif of Gemin5 RNA (previously termed luc-H12) on the 3′ end of the mRNA (Fig. 5a)[20], and the other contains the FMDV IRES on the 5′ UTR (Fig. 5b)[30]. All the Xpress-tagged G5C proteins were expressed at similar levels according to immunoblotting (Fig. 5c, d, Supplementary Fig. 7). The results showed that in comparison with the wild-type G5C (WT), the translation efficiency of the double mutant (L1468D/L1469D) (M1) and the triple (L1381D/M1384D/I1385D) mutant (M2) was significantly repressed by ~1.5–1.8-fold in luc-SL1 and ~1.6–1.5-fold in IRES-luc, respectively. In contrast, the single mutant L1469H (M3) did not impair translation to the same extent (Fig. 5a, b). No significant differences were observed in the mRNA levels determined by RT–qPCR for the different constructs (Supplementary Fig. 8a, b). Therefore, the repression of protein synthesis observed is likely due to the weaker RNA binding affinity of the G5C mutant proteins, which results from the destabilization of the pentamer/decamer. Thus, an intact pentamer/decamer is required for Gemin5-mediated translational regulation in living cells.

## A positively charged surface of G5C coordinates RNA recognition

Given that a stable G5C decamer is necessary for RNA binding, we hypothesized that the high-order assembly of G5C confers its RNA binding capacity and that adjacent G5C dimers likely bind to RNA in a cooperative manner. In turn, this hypothesis suggests that a region outside the pentamer interface could also contribute to RNA binding. Since our previous work indicated that the RBS1 domain of G5C is engaged in RNA binding[22], we examined the potential RNA-binding surface spatially proximal to the RBS1 region.

In the structure of the G5C decamer, two G5C dimers are spaced apart from each other, with the RBS1 of one G5C molecule (molecule C) spatially proximal to the positively charged surface of another molecule (molecule B) (Fig. 6a, b), such that a large positively charged concave surface comprises residues from RBS1$^C$ and TPR$^B$. Several basic residues of TPR$^B$, including R1035, K1061, K1062, and R1090, are spatially close to RBS1$^C$ around the unstructured region between helices α26$^C$ and α27$^C$ (Fig. 6c). Hence, we made a quadruple mutant R1035A/K1061A/K1062A/R1090A (M4) by substituting the four basic residues with Ala (Fig. 6c). M4 behaves as a decamer in the SEC assay (Fig. 6d) but displays a weaker binding affinity toward SL1 RNA (Fig. 6e).

To validate the specificity of the identified concave surface in RNA binding, we made another mutant, K1363A/K1436A/R1437A/R1444A/K1492A (M5), by substituting five basic residues at the pentamer

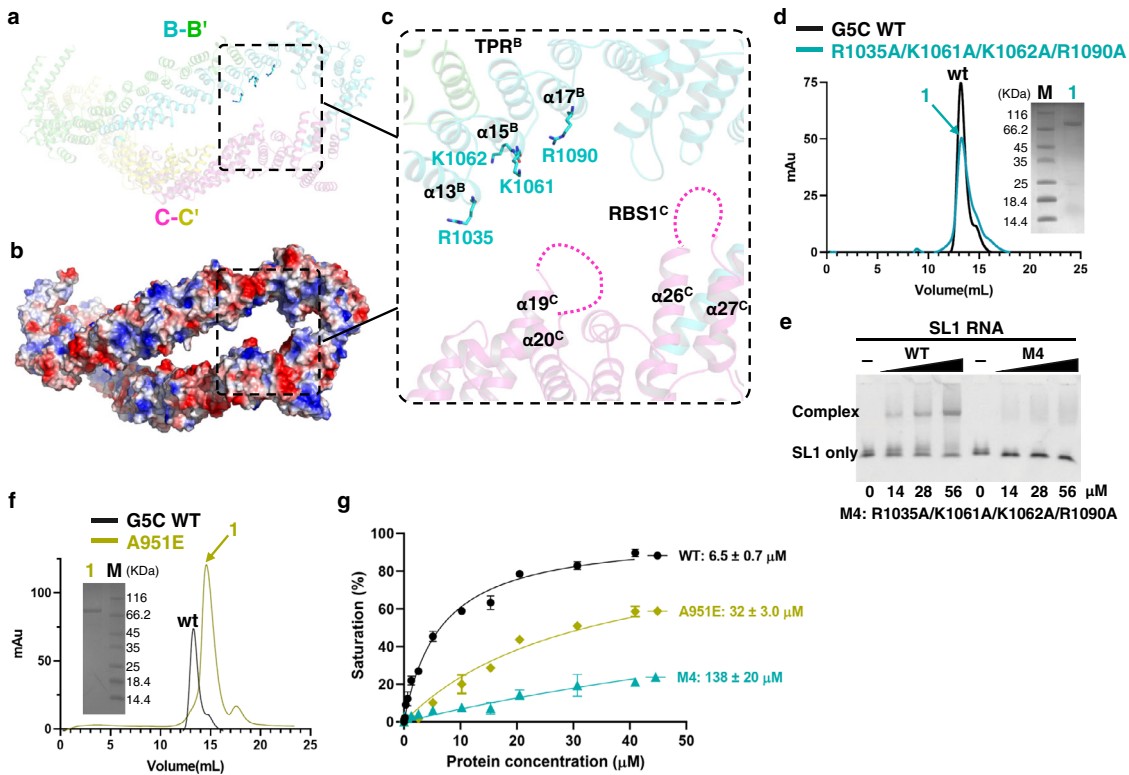

**Fig. 6 | Potential RNA binding surface of the G5C decamer. a** In the structure of the G5C decamer, B–B' and C–C' are two spatially adjacent dimers, with B, B', C, and C' colored cyan, green, pink, and yellow, respectively. **b** Electrostatic surface of the two adjacent dimers, B–B' and C–C'. **c** Close-up view of a large positively charged concave surface as the potential RNA binding site formed by the TPR domain of molecule B (TPR$^B$) and the RBS1 region of molecule C (RBS1$^C$). The four basic TPR residues within α13–α17, including R1035, K1061, K1062, and R1090, are shown in sticks. TPR$^B$ and RBS1$^C$ are indicated. Two invisible loops in molecule C are indicated by dashes. **d** SEC and **e** EMSA for the G5C mutant R1035A/K1061A/K1062A/R1090A. **f** SEC assay for G5C A951E. For Fig. 6d–f, each experiment was repeated independently twice with similar results. **g** FP assay for WT G5C, A951E, and M4 binding to SL1 RNA with the indicated $K_D$ values. Data represented as mean ± SD ($n$ = 2 per concentration and two individual experiments).

interface (within helices 28, 29, and 32) in the opposite orientation from the intermolecular concave surface (Supplementary Fig. 9a–c). In contrast to the quadruple mutant, M5 behaves as an intact decamer and binds to the SL1 RNA only slightly weaker than wild-type G5C (Supplementary Fig. 9d, e), suggesting that these five basic residues are not involved in RNA recognition. Therefore, we conclude that the positively charged concave formed by the TPR dimer and RBS1 structure that includes the unstructured region previously reported to be important for interaction with RNA[22,25], likely serves as the binding site for RNA ligands (Fig. 6b, c).

Our previous study showed that A951E within TPR disrupts the dimer[24]. Consistently, the SEC assay indicated that G5C A951E did not form a decamer (Fig. 6f). Moreover, the FP binding assay also revealed that G5C A951E reduced the SL1 binding affinity by ~5-fold ($K_D$s: 32 vs. 6.5 μM), whereas the M4 mutant weakened the binding affinity by >20-fold ($K_D$s: 138 vs. 6.5 μM) (Fig. 6g), suggesting that an intact pentamer/decamer is required for binding to SL1 RNA. Given that the G5C TPR homodimer alone does not bind to RNA[24], we propose that pentamerization might play a crucial role in positioning two adjacent G5C dimers for cooperatively binding to stem–loop-containing RNA ligands (Fig. 6a, b). In summary, our structural study, complemented by biochemistry and in vivo translation assays, uncovered the molecular basis underlying the G5C decamer and demonstrated that pentamer formation enables the two adjacent dimers to bind RNA ligands in a coordinated manner.

### Structural deficiencies of Gemin5 pathogenic mutations placed in G5C
Recently, biallelic pathogenic mutations identified in Gemin5 were reported to be the basis of neurodevelopmental disorders[11–13,31].

Remarkably, 12 mutations were mapped in G5C, and six of them were found in the TPR domain, including H923P, I988F, S1000P, A1007T, R1016C, and D1019E (Supplementary Fig. 10)[11,31]. While H923P and S1000P disrupt α-helices, I988F, A1007T, R1016C, and D1019E likely introduce steric clashes to destabilize the TPR domain (Supplementary Fig. 10). L1119S, which destabilizes the protein by impairing hydrophobic interactions, are mapped in the G5C MID region. The remaining G5C mutations are located within the pentamer region, including D1264P, Y1282H, Y1286C, Y1286N, and L1367P. D1264P and L1367P would lead to the destruction of the pentamer by destabilizing the protomer, whereas Y1282H, Y1286C, and Y1286N are located near or at the pentamer interface, thereby largely abolishing pentamer formation (Supplementary Fig. 10). Given that the G5C decamer assembly is critical for binding to its cognate RNA ligands, potential loss-of-function mutations in either TPR or the pentamer region would greatly abolish RNA binding by destroying decamer formation. Therefore, we are tempted to suggest that neurodevelopmental disorders mediated by Gemin5 pathogenic mutations placed on its C-terminal region (G5C) are likely associated with aberrant mRNA binding.

## Discussion
As the largest subunit within the SMN complex, Gemin5 is traditionally known for its SMN-dependent function in pre-snRNA recognition and snRNP assembly[7,14,15]. Our previous work identified the RBS1 domain within G5C as a polypeptide with the capacity to interact directly with thermodynamically stable stem-loop regions of viral RNAs and cellular mRNAs[19,21,32]. Residues enabling RNA binding capacity have been identified within the intrinsically unstructured moiety of the RBS domain[22,25]. We also showed that the TPR domain of G5C is a

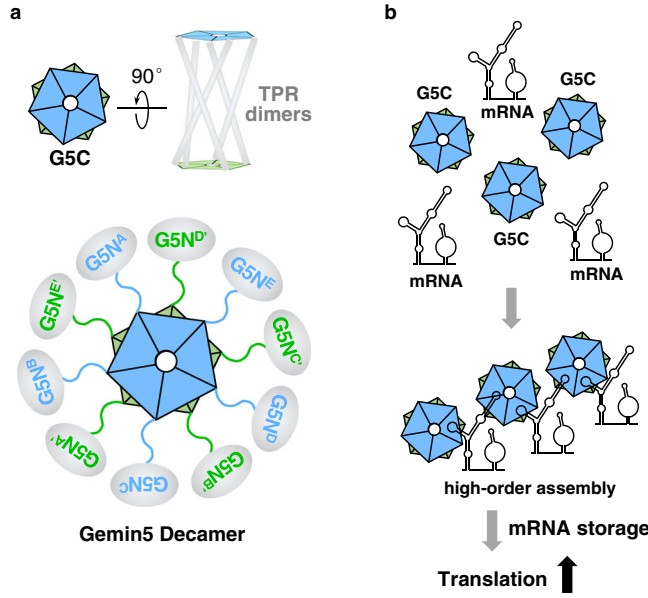

**Fig. 7 | Proposed model for the role of G5C in RNA translation. a** The G5C decamer is a dimer of pentamers, with five TPR dimers bridging two pentamers. Full-length Gemin5 forms a homodecamer via G5C. **b** Multivalent interactions between G5C and stem-loop regions of mRNA enable high-order assembly of RNA–protein complexes. These complexes might upregulate mRNA translation by enabling mRNA storage and preventing mRNA decay.

dimerization module, while other studies reported that purified Gemin5 elutes as a high molecular weight polymer[14,24,33]. However, the molecular mechanism underlying G5C assembly, as well as its RNA binding capacity, is not fully understood because of the lack of the G5C structure as a whole. Here, we show that G5C adopts a homodecameric configuration solely consisting of α helices, with the G5C protomer bearing the TPR dimerization and pentamerization modules at the N- and C-termini, respectively (Fig. 2a). The two most important modules allow G5C to assemble into a compact decamer that can be described as a dimer of pentamers, with five TPR homodimers as arms to connect the two pentamers. Pentamerization-mediated spatial arrangement of the TPR dimers confers G5C RNA binding capacity (Fig. 1b–g).

In the decamer structure, residues of the pentamer interface are highly hydrophobic (Fig. 3d). Our current work validates the pentamer interface by identifying mutations that disrupt the leucine core, thereby destabilizing the assembly of pentamers/decamers. Simultaneously, pentamer-destabilizing mutations in G5C weakened the binding to SL1 RNA (Fig. 4a, b). Therefore, assembly of an intact pentamer/decamer is required for G5C binding to its cognate RNA ligand SL1 and likely other RNA targets. Of note, the mutations impairing pentamer/decamer formation also destabilize the protein conformation, suggesting that pentamer formation plays an important role in G5C stabilization by protecting the evolutionarily conserved hydrophobic surface from the solvent (Supplementary Fig. 6).

From structural analysis, we identified a positively charged surface on the TPR that is ~33 Å apart from RBS1, a previously identified RNA binding region within G5C (Fig. 6a–c)[22]. Mutations of four basic residues within the TPR domain near the RBS1 moiety did not alter the decamer architecture but weakened the binding to SL1 (Fig. 6d, e). Given that the TPR module alone does not bind RNA[24], we propose that two adjacent protomers are coordinated after pentamerization to bind SL1 RNA. In this way, the SL1 RNA could be contacted by the TPR from one protomer and the RBS1 from the other. The SL1 RNA is predicted to have a size of ~70 Å, thereby large enough to form a bridge between both protomers (Fig. 6a–c). This hypothesis also accounts for the observation that G5C binds preferentially to RNAs containing long stem loops[20].

Interestingly, Gemin5 downregulates viral RNA translation[19], while it promotes translation of its own mRNA in vivo[20]. The newly discovered G5C decamer structure allows us to propose a model to solve the apparently opposite roles of Gemin5 in translation. Full-length Gemin5 forms a homodecamer via G5C, which is connected with G5N via an intrinsically disordered linker (Fig. 7a), although how G5N is placed in the overall structure remains to be studied in future studies. Thus, independent of the role of G5N in SMN complex assembly, the assembly of G5C into a decamer structure protects RNA ligands from decay by binding to their thermodynamically stable stem-loop regions to form protein–RNA complexes (Fig. 7b). In addition, it was reported that during the stress response, Gemin5 is recruited into the cytoplasmic granule response[34–36], which might facilitate its role in the storage of mRNAs[37,38]. Hence, in contrast to G5N, which exhibits strict sequence specificity[14], G5C binds preferentially to stem-loop regions of RNA ligands[20], depending on RNA secondary structures rather than sequence. Indeed, a supershift band was observed in the EMSA for G5C binding to SL1, in full agreement with earlier results[22], suggesting the formation of high-order complexes (Fig. 1c). The weaker interaction between G5C and SL1 or other cognate RNA ligands, as well as G5C pentamerization, might trigger the formation of granules, as observed for other protein–RNA complexes[39,40]. Previous work also suggested the role of Gemin5 associated with the P-body in RNA decay[14,34]. Therefore, the exact role of Gemin5 in RNA translation might depend upon the RNA target and the cellular conditions.

In summary, our current study provides near-atomic structural information on the C-terminal region of Gemin5, a missing knowledge of this essential protein necessary to interpret its various functions in RNA-related processes. We also demonstrated the potential impact of human pathogenic mutations recently reported in the *Gemin5* gene on the tertiary structure of the G5C protein. Future studies will be required to examine the role of the G5C–RNA interaction in stress granule formation and translation regulation and to expand this information to the full-length protein.

## Methods

### Protein expression and purification

The gene encoding the Gemin5 C-terminal fragment (Gemin5$_{841-1508}$) was amplified by PCR from a cDNA library and cloned into a modified pET28a vector fused with an N-terminal hexa-histidine tag. The recombinant protein was overexpressed in *Escherichia coli* BL21 (DE3). Cells were grown in LB medium at 37 °C until the OD$_{600}$ reached -0.6. Protein expression was induced with 0.5 mM (final concentration) β-D-1-thiogalactopyranoside (IPTG) for 20 h at 16 °C. Cells were harvested by centrifugation at 3600 × *g* for 10 min at 4 °C. Pellets were resuspended in lysis buffer containing 20 mM Tris, pH 7.5, 400 mM NaCl, and 5 mM imidazole. Recombinant proteins were purified by Ni-NTA (GE Healthcare). After washing with buffer containing 20 mM Tris, pH 7.5, 400 mM NaCl, and 20 mM imidazole, the proteins were eluted with 20 mM Tris, pH 7.5, 400 mM NaCl, and 250–500 mM imidazole. After elution, recombinant Gemin5$_{841-1508}$ was treated with TEV protease overnight to remove the N-terminal His-tag. Then, the cleaved recombinant proteins were further purified by Superdex 200 gel filtration and mono Q ion exchange (GE Healthcare). Purified protein was concentrated at 1.5 mg/ml in a buffer containing 20 mM Tris–HCl (pH 7.5), 300 mM NaCl, and 0.5 mM DTT for future use, including cryo-EM sample preparation.

Site-specific mutations were performed using two reverse and complement primers containing the mutation codon. The primer sets used for mutations are listed in Supplementary Table 3. All G5C mutants were purified in the same way as wild-type G5C.

### Multiangle static light scattering

The molecular mass analysis of wild-type Gemin5 was performed on an AKTA Pure system (GE Healthcare) coupled with a DAWN HELEOS 8+

instrument (Wyatt Technology). One hundred microliters of wild-type Gemin5 protein samples (1 mg/ml) were loaded into a Superose 6 Increase 10/300 GL column (GE Healthcare) pre-equilibrated by a buffer composed of 20 mM Tris−HCl, pH 7.5, and 330 mM NaCl. The data were analyzed with ASTRA software (Wyatt).

### RNA preparation

The RNAs used for EMSA were derived from the stem0loop SL1 (nt 18–102) of H12 RNA[22] and the stem-loop (nt 416–462) of FMDV IRES RNA[21] that were transcribed and purified in vitro as described previously[14]. The synthesized DNA template (Sangon Biotech.) was amplified by PCR before being used for tRNA transcription. Then, the amplified DNA templates were purified by isopropanol precipitation and dissolved in diethyl pyrocarbonate (DEPC)-treated water. The 20 µl in vitro transcription mixture contained 2 U TranscriptAid Enzyme Mix (Thermo Scientific TranscriptAid T7 High Yield Transcription kit), 4 µl 5× TranscriptAid buffer, 3.5 µM DNA template, 10 mM NTPs, and DEPC-treated water (Thermo Fisher Scientific Kit). The mixture was incubated at 37 °C for 8 h. After transcription, 2 µl DNase I from the kit was added to the mixture, and the mixture was further incubated at 37 °C for 0.5 h to remove the DNA template.

Each 60 µl of transcription product was treated with 500 µl of RNAiso Plus, shaken for 15 min, and then mixed with 100 µl of chloroform. The mixture was centrifuged at $14,000 \times g$ for 15 min. The supernatant was collected and further purified by isopropanol precipitation and ethanol precipitation methods. After being dissolved into DEPC-treated water, RNA was further purified by HiTrapTM Q HP (GE Healthcare). The purified RNA was annealed to generate folded RNA before further use.

### RNA electrophoretic mobility shift assay (EMSA)

RNA-binding reactions were carried out in 10 µl of RNA-binding buffer (100 mM NaCl, 20 mM Tris−HCl pH 7.5) for 1 h on ice. Increasing amounts of protein were incubated with a constant concentration of SL1 RNA (0.35 µM) or IRES RNA (0.35 µM). Electrophoresis was performed in native 3.0% (19:1) polyacrylamide gels. The gels were run at 110 V for 30 min in 0.5× TBE (Tris/borate) buffer made from a 10× TBE stock solution. Then, the gel was stained by GelRed staining, and the images were processed by ImageJ software[41]. All shifted bands, including the upper bands, were considered for $[RNA]_{Bound}$, and the fraction bound value was defined as $[RNA]_{Bound}/([RNA]_{Unbound} + [RNA]_{Bound})$. The curves were analyzed with GraphPad Prism 8. Quantitation data from the EMSA for wild-type G5C binding to SL1 and IRES RNAs are shown in Supplementary Table 4. Original gels are shown in Supplementary Fig. 11.

### Fluorescence polarization assay

Fluorescence polarization assays were performed with purified G5C WT and its mutants. All RNAs used for the FP assay were labeled with a 5′ 6-FAM group (Beyotime Biotechnology). Experiments were performed in buffer (20 mM Tris−HCl, pH 7.5, 150 mM NaCl). Each well contained 10 nM RNA (40 nM for poly U) and different protein concentrations (in a range of 0−41 µM) in a total volume of 80 µL. We used black flat bottom 384-well plates (Corning, 3571) and a CLARIOstar Grating Multi-Microplate Reader for data reading. The excitation and emission wavelengths were 485 and 520 nm, respectively. The dissociation equilibrium constant $K_D$s were obtained by fitting the saturation (%) with protein concentrations. The curve fitting was performed by GraphPad Prism 8.

### Cryo-EM sample preparation, data acquisition, and data processing

Three microliters of sample were applied onto glow-discharged 200-mesh R2/1 Quantifoil Au grids. The grids were blotted for 3.5 s in 100% humidity at 8 °C with no blotting offset and rapidly frozen in liquid ethane using a Vitrobot Mark IV (Thermo Fisher).

The G5C grids were screened using a Talos Arctica cryo-electron microscope (Thermo Fisher Scientific) operated at 200 kV. Good grids were then imaged in a Titan Krios cryo-electron microscope (Thermo Fisher Scientific) with a GIF energy filter (Gatan). Micrographs were recorded in superresolution mode with a pixel size of 0.535 Å at a dose rate of 8 e−/pixel/s. Each image was composed of 40 individual frames with an exposure time of 2.5 s. A total of 4888 movie stacks were collected in super-resolution mode with a K3 camera at a nominal magnification of ×81,000 with a defocus range from −2.5 to −1.5 µm.

### Image processing

MotionCor2[42] was used for motion correction and dose weighting. CTFFIND4[43] was used for the contrast transfer function estimation. A total of 3,126,329 particles were autopicked and extracted in CryoSPARC[44], and then extracted particles were subjected to 2D classification with good classes selected for 3D classification. A total of 1,577,779 particles were used for Ab initio 3D reconstruction in CryoSPARC into three classes. Then, the best class containing 870,934 particles was selected for further homogeneous refinement, generating a map of 3.79 Å resolution. Next, nonuniform refinement together with local and global CTF refinement was performed with D5 symmetry imposed, yielding a map with 3.3-Å resolution. To obtain a better structure of the protomer, symmetry (D5) expansion was used, increasing the particle number by 10 times. The final map of the protomer was achieved by local refinement with a resolution of 2.6 Å. Map resolution was estimated by the "gold standard" Fourier shell correlation (FSC) at the 0.143 criterion. Local resolutions were estimated using the Local Map Estimation program in CryoSPARC[44], with the local resolution map depicted by UCSF Chimera[45].

### Model building and refinement

The final sharpened map with a B-factor of −150 Å² was used for model building in Coot[46]. By using PHENIX map-to-model[47], the solved G5C TPR domain structure (PDB ID: 6RNQ)[24] was docked into the cryo-EM map. For the rest of G5C, the predicted model from alphafold (https://alphafold.ebi.ac.uk/) was divided into several fragments and used in model building guided by bulky residues, such as Tyr, Phe, Arg, etc. Manual refinement was performed to remove invisible G5C fragments and to build fragments into the cryo-EM map by Coot[46]. The final structure contains residues 847–1132, 1155–1293, 1346–1391, and 1430–1496. Structure refinement was carried out by using PHENIX[47]. PyMOL (https://pymol.org/) and UCSF Chimera[45] were used for figure preparation.

### Translation assays

The plasmid pcDNA3-Xpress-G5$_{845-1508}$ was previously described[48]. Constructs pcDNA3-Xpress-G5$_{845-1508}$-L1469H, pcDNA3-Xpress-G5$_{845-1508}$-L1468D/L1469D, and pcDNA3-Xpress-G5$_{845-1508}$-L1381D/M1384D/I1385D were generated by QuickChange mutagenesis (Agilent Technologies) using specific primers (Supplementary Table 2). All plasmids were confirmed by DNA sequencing (Macrogen).

HEK293 cells were cultured in Falcon© six-well plates with Dulbecco's modified Eagle's medium (DMEM) supplemented with 5% (v/v) fetal bovine serum (FBS). Cell monolayers (2 × 10⁵) were cotransfected as described[20] using a plasmid expressing luciferase in a cap-dependent or IRES-dependent manner (luc-SL1, pIRES-luc)[49] and a plasmid expressing Xpress-G5$_{845-1508}$, Xpress-G5$_{845-1508}$-L1469H, Xpress-G5$_{845-1508}$-L1468D/L1469D, Xpress-G5$_{845-1508}$-L1381D/M1384D/I1385D or the empty vector side by side using Lipofectamine LTX (Thermo Fisher Scientific). Cell lysates were prepared 24 h post-transfection in 150 µl lysis buffer (50 mM Tris−HCl pH 7.8, 100 mM NaCl, 0.5% NP40). Luciferase activity (RLU)/µg of total protein was

internally normalized to the value obtained with Xpress-G5$_{845-1508}$ performed side by side. Each experiment was repeated independently three times. Values represent the mean ± SD. We computed $P$ values for a difference in distribution between two samples with the unpaired two-tailed Student's $t$ test. Differences were considered significant when $P < 0.05$. The resulting $P$ values are graphically illustrated in figures with asterisks as described in the figure legends.

## Immunodetection

The protein concentration in the lysate was determined by the Bradford assay. Equal amounts of protein were loaded in SDS–PAGE and processed for western blotting to determine the expression of the polypeptides using anti-Xpress antibody (Thermo Fisher Scientific, catalog R910-25, monoclonal, lot 2190234). The Clone name for Invitrogen anti-Xpress is reference 46-0528 (lot 2190234). RRID of mouse monoclonal IgG1 is AB_2556552. Immunodetection of tubulin (Merck) was used as a loading control. The anti-tubulin antibody Sigma is monoclonal DM1A (ascites fluid) (catalog T9026, lot 096k4777). Goat anti-mouse (H + L) secondary antibody from Invitrogen (Thermo Fisher Scientific, catalog 32430, lot VJ313743) was used according to the manufacturer's instructions. The signal detected was performed in the linear range of the antibodies. The dilutions for anti-Xpress and anti-tubulin are 1:2000 and 1:4000, respectively. The dilution for the secondary antibody is 1:2000.

## RNA quantification

To measure the mRNA steady-state levels, total RNA was isolated from lysates prepared from cells harvested 24 hpt, expressing the corresponding plasmids using TRIzol (Thermo Fisher Scientific), precipitated with isopropanol, and resuspended in RNase-free $H_2O$. Reverse transcriptase (RT) reaction was performed to synthesize cDNA from equal amounts of the purified total RNA samples using SuperScriptIII (Thermo Fisher Scientific) and hexanucleotide mix (Merck) as primers. For quantitative PCR (qPCR), the oligonucleotides 5'Luciferase/3'Luciferase[20] and Xpress-s/Xpress-as[29] were used. qPCR was carried out using the NZYSupreme qPCR Green Master Mix (NZytech) according to the manufacturer's instructions on a CFX-384 Fast Real-time PCR system (Bio-Rad). Values were normalized against constitutive MYO5A RNA[20]. The comparative cycle threshold method[27] was used to quantify the results.

## Reporting summary

Further information on research design is available in the Nature Research Reporting Summary linked to this article.

## Data availability

The cryo-EM structures of the G5C decamer and protomer are deposited into the protein data bank (PDB) with the accession numbers 7XDT, and 7XGR, respectively. The cryo-EM density maps are deposited in the Electron Microscopy Data Bank (EMDB) under accession numbers EMD-33152 and EMD-33187. All other data supporting this study are available within the paper and its Supplementary Information file. Source data are provided with this paper.

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

## Acknowledgements

We thank the Cryo-EM Center at the University of Science and Technology of China for the support of cryo-EM data collection. We thank Dr. Yong-Xiang Gao and the Cryo-EM Center at the University of Science and Technology of China for technical support on cryo-EM data collection. This work is supported by the National Natural Science Foundation of China Grants (22137007 and 92053107 to C.X.) and Ministerio of Science and Education of Spain (grant PID2020-115096RB-I00 to EMS). C.X. is supported by the Fundamental Research Funds for the Central Universities, "the Thousand Young Talent program", and the Major/Innovative Program of Development Foundation of Hefei Center for Physical Science and Technology (2021HSC-CIP014).

## Author contributions

C.X. and Q.G. conceived the project. Q.G. and S.Z. performed structural biology and biochemical experiments with assistance from J.Z., P.T., H.S., and L.S.; K.Z. and Q.G. performed cryo-EM experiments; R.F.-V., A.E.-B., and S.A. performed the in vivo translation assay; S.Z., M.L., and Q.G. performed the FP binding assays and analyzed the data; X.Y., J.M., and Y.S. provided reagents and the assistance in biochemical assays; C.X., K.Z., and E.M.-S. wrote the manuscript with input from all authors; C.X., K.Z., and E.M.-S. supervised the project.

## Competing interests

The authors declare no competing interests.
