## [Peer Review File · Nature Communications]

Structural basis for Gemin5 decamer-mediated mRNA bindingREVIEWER COMMENTS

Reviewer #1 (Remarks to the Author):

This is an interesting manuscript describing the cryo-EM structure of the C-terminal half of Gemin5 and mapping of its RNA binding through mutagenesis. The structure is quite interesting - it is a dimer of pentamer (or vice versa) and the authors suggest a model of how this oligomeric structure can support granule formation by polymerizing with RNAs. They also analyse pathogenic mutations in disease, and performed mutagenesis to support the oligomerization model. The figures are clear and informative.

The paper is well written and succinct. I support publication.

Reviewer #2 (Remarks to the Author):

In this manuscript, Guo et al reports the structural analysis of Gemin5 C-terminal region (Gemin5C) using cryo-EM and shows the role of this specific region in oligomerization for a decamer formation, mRNA-binding and translational regulation. They further show the relevance of Gemin5C high-order structure in Gemin5 mutation-driven pathogenic mechanisms and suggest the role of Gemin5 in the regulation of RNA metabolism.

Gemin5 has been indicated to regulate translation on select mRNAs via direct binding to RNA. However, there are contradictory reports on whether Gemin5 has a positive or negative regulation on translation. The model proposed in this study shed a new light on the mechanistic insights of how this regulation might occur. Importantly, the results shown in this study provide a high resolution of the structural organization of Gemin5C. Strikingly, Gemin5C forms a pentagon-like structure and two units of this oligomer form a decamer through the previously proposed model of TPR dimerization. The structural findings reported in this manuscript are surprising and completes a molecular architecture of the Gemin5 functions. However, the significance of structural analysis and the conclusions made from this study are not supported by the current biochemical data. In fact, most, if not all, of biochemical data in the manuscript are in the primitive stage and lack good negative and positive controls. The authors should address these major concerns in their revision.

Major points)

1. The authors used truncated Gemin5C for the entire study. Considering the deletion of N-terminus half of the Gemin5 protein, the structural organization of Gemin5C may only exist when the Gemin5 protein gets truncated. The authors should address whether the structure of Gemin5C still exists in the full-length context using biochemical analysis. This reviewer is NOT asking to get the cryo-EM structure of full-length Gemin5. Any indirect or direct biochemical evidence that this high-ordered structure of Gemin5C still exists in the full-length Gemin5 protein would be good.

2. One of the critical readout experiments that authors utilized for structure-function relationship is EMSA. However, the entire data using EMSA in the manuscript are not in high quality and raise many questions regarding the rigor of the data, transparency of data presentations and conclusions. Overall, biochemical data for Gemin5C-RNA interaction need improvements. It is highly recommended to use a better quantitative methodology to measure the affinity.

- Conditions of EMSA: In materials and methods, the authors stated the experimental condition of EMSA. The condition they used is not acceptable. In the condition like 100 mM NaCl and 20 mM Tris/HCl 7.0, it is highly likely that any non-specific interactions contribute to the RNA binding, especially when a recombinant protein is used for EMSA in a defined simple binding environment. In typical assay conditions, a cold competitor tRNA is used along with a detergent and a higher salt concentration.

- Data interpretations: This is the part that this reviewer is struggling a lot. First, the EMSA data in Figure 1g did not even reach to the saturation in the binding reaction. So, it is not clear to me how the authors still could calculate the Kd. Second, Gemin5C-RNA probe in gels should be validated by supershift experiments. SL1 RNA is twice bigger than IRES (FMDV) RNA in their size but the Gemin5C-SL1 RNA bands are much smaller than Gemin5C-IRES (FMDV) RNA in gels. Besides, SL1 RNA-Gemin5C EMSA data show a upshifted band which only appears after the addition of Gemin5C, indicating that this band appearance is also coming from the existence of Gemin5C. Despite this fact, it seemed that the authors did not pay attention to this upper band, whose existence likely affects the calculation of the affinity. Thus, the authors should clarify the issue with this upshifted band using SL1 as a probe.

- Rigor of data: The authors used a wrong statistic parameter. Experimental results like EMSA with multiple biological repeats should be presented using SD not SEM. SEM always minimizes the error bar. Besides, the authors should note that the dispersion of the data from the mean (SD) is of interest but not the quantitation of uncertainty in estimating the mean (SEM) in presenting data with biological repeats. The authors should also provide the images of their other biological repeat experiments as supplementary data. A table of quantitation from EMSA images would be also helpful to see how they calculated the band intensity and converted the information into the quantity of the RNA probe.

- Inconsistency in the affinity: A previous study showed that the RBS1 domain (part of Gemin5C) has a low to mid nanomolar range affinity to FMDV IRES. In this study, the affinity to the same RNA is around 50 μ M. This is very difficult to understand unless the formation of Gemin5C decamer actually affects the affinity to RNA. This difference also brings back the question of whether a high-order decameric structure is naturally existing structure or not and the EMSA experiments were done properly.

- Specificity of the binding: Based on the argument made by the authors, Gemin5N specifically binds to the snRNP code of snRNAs. Thus, ideally, Gemin5N should be used as a negative control for at least first set of EMSA experiments.

3. The luciferase assay results in Figure 5 should be presented as SD not SEM. SEM tends to minimize the error bar and thus can make insignificant data to be statistically significant. Please also put the actually p-value instead of stating $p < 0.05$.

4. The experimental design for luciferase assay lacks many controls. First, the authors should show how they controlled the transfection efficiency of both Gemin5 and its mutants expression and the luciferase plasmid. The current format only provides the expression level of Gemin5 and its mutants by western, which is pretty much eyeballing but not quantitative at all. The authors also should show western blot images for each biological repeat. More importantly, it is not clear how the authors are sure about the same transfection efficiency of luciferase plasmid in each Gemin5 and its mutants transfection. Another control missing in the experiment is the luciferase activity without Gemin5C binding domain. The fold changes in this type of experiments are misleading as they tend to hide a magnitude of data point fluctuation by the nature of sensitive assay. It would be better to show the normalized luciferase activity in the arbitrary unit (AU).

5. The results shown in Figure 6 are intriguing, but they need more thoughtful interpretations. Why did the mutations in the TPR residues of the pocket affect the RNA binding while the mutations of RBS1 residues in the same pocket have no impact on RNA binding? Particularly, multiple studies reported RBS1 as an RNA-binding module in Gemin5C. Thus, the authors should explain the discrepancy between their findings and others.

6. In their model shown in Figure 7c, the authors used PTB as an example. Ironically, this model does not consider the affinity of RNA binding proteins to their target RNA at all. PTB is known to be specific

to its target sequences with a high affinity. It is almost impossible for mid to high uM affinity interaction (Gemin5C to target RNA binding) can compete with such a high affinity interaction made by PTB. Either the authors should revise their model or should provide evidence that Gemin5C indeed can outcompete the PTB protein for their RNA binding.

Minor points)

1. There are incomplete sentences in the manuscript.
2. In supplementary Figure 5, the second sequence alignment from the bottom has a wrong labeling on Gallus gallus. Other alignments used Gg but the second one from the bottom alignment used Ch (Chicken).

Reviewer #3 (Remarks to the Author):

In this interesting manuscript, Guo et al. report the three-dimensional structure of the C-terminal region of Gemin5 (G5C). Gemin5 is an essential component of the SMN complex, an RNA-binding protein involved in the formation of small nuclear ribonucleoproteins. It is shown that G5C adopts a homodecameric structure assembled from a dimer of pentamers. The structure was thoroughly validated using mutagenesis together with in vitro and in cell assays. It is convincingly shown that a large positively charged concave surface brought about by G5C oligomerization is needed for the interaction of Gemin5 with stem-loop-containing RNA molecules. Based on the G5C structure, the authors make a good case that some of the previously identified pathogenic mutations in Gemin5 probably prevent pentamer formation and RNA binding. In conclusion, Guo et al. have uncovered a unique oligomeric arrangement of G5C that is key for binding stem-loop regions of RNAs. The new structural knowledge resulting from this carefully done study provides a rigorous framework for testing hypotheses related to the functions and modes of action of Gemin5 in SMN. This work is of high technical quality and is appropriate for publication in Nature Communications.

Minor points:

SL1 (stem-loop 1) should be defined in the text.

In Supplementary Table 1, an FSC threshold of 0.125 is indicated for the Gemin5 (841-1508) protomer. Where does this value come from? Is it a typo?

Reviewer #4 (Remarks to the Author):

The authors determined the structure of Gemin5 C-terminal domain that behaves as a decamer. Previously, it has been known that Gemin5 N-terminal WD40 recognizes snRNAs to ensure the assembly of snRNPs. This study dissects the mRNA binding specificity of Gemin5 CT. They uncovered the molecular mechanism underlying decamer formation and examined the potential mRNA binding interface on Gemin5 CT. Overall, the cryo-EM structure is of high quality and these data must be valuable for the researchers in the RNA field. However, some issues should be addressed before it could be accepted for publication.

1. Is that structure similar to any solved structures? Can they compare the Gemin5 decamer with other similar modules?
2. The authors examined the binding between Gemin5 and SL1, and Gemin5 and IRES. The sequences of the two RNAs are quite different. Could the authors further generate some RNA variants to examine whether the binding is based on the RNA structure, such as the length of stem loop?
3. There are some loops invisible in the cryo-EM structure of Gemin5, it is not clear whether they also play roles in RNA binding.

4. For the cryo-EM map reconstitution and refinement, there should be an FSCwork/FSCfree curve to show the absence of overfitting in model refinement. A model vs map curve should also be reported.

Revision summary

We are very grateful to the reviewers for their thoughtful comments and suggestions that have tremendously helped us improve the quality of our current manuscript. All four reviewers acknowledged the significance of our work revealing the novel high order structure of GEMIN5 C-terminal half region. We strongly believe that we have appropriately addressed the issues raised by the reviewers. We are providing a point-to-point response to the comments below. For the convenience of examination, we have highlighted all the changes in red in the revised text.

Reviewer #1 (Remarks to the Author):

This is an interesting manuscript describing the cryo-EM structure of the C-terminal half of Gemin5 and mapping of its RNA binding through mutagenesis. The structure is quite interesting - it is a dimer of pentamer (or vice versa) and the authors suggest a model of how this oligomeric structure can support granule formation by polymerizing with RNAs. They also analyse pathogenic mutations in disease, and performed mutagenesis to support the oligomerization model. The figures are clear and informative.

The paper is well written and succinct. I support publication.

Response: We are grateful to the reviewer for his/her positive view of our manuscript, and also for commenting on the importance and significance of our current work. We appreciate your kind words on the organization of our manuscript.

Reviewer #2 (Remarks to the Author):

In this manuscript, Guo et al reports the structural analysis of Gemin5 C-terminal region (Gemin5C) using cryo-EM and shows the role of this specific region in oligomerization for a decamer formation, mRNA-binding and translational regulation. They further show the relevance of Gemin5C high-order structure in Gemin5 mutation-driven pathogenic mechanisms and suggest the role of Gemin5 in the regulation of RNA metabolism.

Gemin5 has been indicated to regulate translation on select mRNAs via direct binding to RNA. However, there are contradictory reports on whether Gemin5 has a positive or negative regulation on translation. The model proposed in this study shed a new light on the mechanistic insights of how this regulation might occur. Importantly, the results shown in this study provide a high resolution of the structural organization of Gemin5C. Strikingly, Gemin5C forms a pentagon-like structure and two units of this oligomer form a decamer through the previously proposed model of TPR dimerization. The structural findings reported in this manuscript are surprising and completes a molecular architecture of the Gemin5 functions. However, the significance of structural analysis and the conclusions made from this study are not supported by the current biochemical data. In fact, most, if not all, of biochemical data in the manuscript

are in the primitive stage and lack good negative and positive controls. The authors should address these major concerns in their revision.

Response: We thank the reviewer for his/her positive comments on our study. We have translated his/her great input and specific points into a well-versed revision.

Major points)

1. The authors used truncated Gemin5C for the entire study. Considering the deletion of N-terminus half of the Gemin5 protein, the structural organization of Gemin5C may only exist when the Gemin5 protein gets truncated. The authors should address whether the structure of Gemin5C still exists in the full-length context using biochemical analysis. This reviewer is NOT asking to get the cryo-EM structure of full-length Gemin5. Any indirect or direct biochemical evidence that this high-ordered structure of Gemin5C still exists in the full-length Gemin5 protein would be good.

Response: Thanks for this comment. Obtaining the structural information of the full-length protein is amongst our future objectives. However, as the reviewer may have anticipated, this requires far more work, and it is beyond the scope of this manuscript. Obtaining the full-length (FL) Gemin5 in sufficient concentration and purity is currently a main obstacle to carry out experimental approaches showing, or not, the assembly of a high order protein structure. Currently there is no direct evidences of how the first half containing the WD repeats domain (**PMID: 27881600**) and the last half of the protein (this manuscript) are interconnected. Nevertheless, biochemical data obtained using cell lysates suggest that the protein forms oligomers. Data in favor of the oligomerization of Gemin5 is the observation that overexpression of G5₁₋₁₂₈₇ recruits the endogenous protein, therefore forming at least a dimer (**PMID: 27507887**).

Additional data indicates that for the recruitment to occur it is necessary the integrity of the TPR module as the dimerization domain (**PMID: 31799608**) and the integrity of the RBS1 domain (**Francisco Velilla et al., unpublished**), suggesting that assembly of the FL-Gemin5 protein into a high-order macromolecular complex is at least in part determined by the structural organization of the C-terminal half. In addition, the protein purified from insect cells forms a large macromolecular complex (compatible with a polymer in gel filtration, data not shown in **PMID: 26069323**). Our attempts to obtain further biochemical evidences of protein oligomerization indicated that the CT part of the protein recruits the endogenous counterpart forming oligomers detected by mass spectrometry, although the MW of these complexes is not known. Attempts to determine the assembly of large MW protein aggregates in native gels failed, as they form a smear. Together, these preliminary observations suggest the assembly of a high order structure though further structural studies will be needed in the future.

2. One of the critical readout experiments that authors utilized for structure-function relationship is EMSA. However, the entire data using EMSA in the manuscript are not in high quality and raise many questions regarding the rigor of the data, transparency of data presentations and conclusions. Overall, biochemical data for Gemin5C-RNA

interaction need improvements. It is highly recommended to use a better quantitative methodology to measure the affinity.

Response: We thank the reviewer for this suggestion. We removed the K_D values from EMSA assay and only showed the curves from duplicate EMSA experiments (**revised Fig. 1f**). The maximum G5C concentration is $\sim 50 \mu\text{M}$, and it is not likely to obtain reliable K_D for G5C binding to IRES. We used FP binding assay to measure the K_D for G5C binding with SL1 (**revised Fig. 1g**). Given that G5C likely prefers stem-loop-containing RNAs, we used FAM-labeled shorter form of SL1 (SL1^{short}) and FAM-labeled single stranded poly-U RNA (U₇) as the control. The FP binding assay demonstrates that G5C binds to SL1 and SL1^{short} with K_D s of $6.5 \mu\text{M}$ and $67 \mu\text{M}$, respectively. In contrast G5C displays no binding towards the single stranded RNA U₇ ($K_D > 500 \mu\text{M}$). Therefore, G5C preferentially binds to SL1 over its truncated form or short ssRNA.

- Conditions of EMSA: In materials and methods, the authors stated the experimental condition of EMSA. The condition they used is not acceptable. In the condition like 100 mM NaCl and 20 mM Tris/HCl 7.0, it is highly likely that any non-specific interactions contribute to the RNA binding, especially when a recombinant protein is used for EMSA in a defined simple binding environment. In typical assay conditions, a cold competitor tRNA is used along with a detergent and a higher salt concentration.

Response: As the reviewer suggested, we would like to show the fraction bound only and remove the K_D values (**revised Fig. 1f**). To capture enough protein-RNA complex in EMSA, it is not rare to use lower salt concentration in EMSA assay. For example, 50 mM KCl (**PMID: 24284625, 25242552**), 100 mM NaCl (**PMID: 27881601**), 100 mM KCl (**PMID: 35301482**) or even lower salt concentration (**PMID: 27628236**), were used for studies on protein-RNA interactions. For low affinity protein-RNA complex, it is difficult to saturate RNA with protein even by using fluorescently labeled RNA in a binding assay with 50 mM KCl salt concentration (**PMID: 34413138**).

In previous assays we have shown that purified proteins have distinct RNA-binding capacity, as follows: 1) the RBS1 domain does not interact with short probes (such as UUUCCUUU) under the same conditions used for SL1 binding (**PMID: 32476560**); 2) the purified TPR domain does not interact with any of the numerous probes used (D5, a stable hairpin of 22 nt, a long RNA of 180 nt, a ssRNA of 32 nt, among others) (**PMID: 31799608**).

However, we still followed the reviewer's suggestion by increasing salt concentration. We used buffer containing 20mM Tris pH 7.5, 150 mM NaCl in the FP binding assay with FAM-labeled single stranded RNA U₇ as the negative control. G5C decamer selectively binds to SL1 but not U₇ (see new **Fig 1g**).

- Data interpretations: This is the part that this reviewer is struggling a lot. First, the EMSA data in Figure 1g did not even reach to the saturation in the binding reaction. So, it is not clear to me how the authors still could calculate the Kd. Second, Gemin5C-RNA probe in gels should be validated by supershift experiments. SL1 RNA is twice bigger than IRES (FMDV) RNA in their size but the Gemin5C-SL1 RNA bands

are much smaller than Gemin5C-IRES (FMDV) RNA in gels. Besides, SL1 RNA-Gemin5C EMSA data show a upshifted band which only appears after the addition of Gemin5C, indicating that this band appearance is also coming from the existence of Gemin5C. Despite this fact, it seemed that the authors did not pay attention to this upper band, whose existence likely affects the calculation of the affinity. Thus, the authors should clarify the issue with this upshifted band using SL1 as a probe.

Response: Thanks for this comment. Previous assays conducted with the RBS1 domain (aa 1287-1412) demonstrated the presence of a non-canonical RNA binding site in the protein. These studies however showed significant differences in the interaction with different probes, the SL1 motif of H12 RNA, and D5 of the FMDV IRES. Four retarded complexes are observed in the case of SL1 RNA (PMID: 32476560), while only 2 are observed in the case of D5 (PMID: 34424823). The proteins used in this manuscript are different and also are the labelled RNAs. For EMSA assay, we consider that the up-shifted band in Fig. 1c is part of the retarded complex (See below).

As the reviewer suggested, we did not use the EMSA assay to obtain K_D values, and removed the K_D value and only show fraction bound (revised Fig. 1f). The unsaturated situations were also observed due to weak binding affinity in previous literature (PMID: 34413138). For EMSA, all clearly shifted bands, including the upper bands, were already considered as the part of the retarded complex. The fraction bound value was defined as $[RNA]_{Bound} / ([RNA]_{Unbound} + [RNA]_{Bound})$. We used FP assay to obtain K_D values for G5C binding with different RNAs. We also added it in the method.

- Rigor of data: The authors used a wrong statistic parameter. Experimental results like EMSA with multiple biological repeats should be presented using SD not SEM. SEM always minimizes the error bar. Besides, the authors should note that the dispersion of the data from the mean (SD) is of interest but not the quantitation of uncertainty in estimating the mean (SEM) in presenting data with biological repeats. The authors should also provide the images of their other biological repeat experiments as supplementary data. A table of quantitation from EMSA images would be also helpful to see how they calculated the band intensity and converted the information into the quantity of the RNA probe.

Response: We have modified the presentation of the data, showing SD in all cases, and removed K_D values in revised Fig. 1f. We have modified the presentation of the data by showing two curves in revised Fig. 1f. We also showed original gel figures for duplicate experiments (Fig. 1f) as Supplementary Fig. 11. A table of quantitation from EMSA images was added as Supplementary Table 4. Additionally, we have added a new experiment in Fig. 1g showing K_D s measured by FP, as mentioned above. We hope the reviewer is satisfied with that.

- Inconsistency in the affinity: A previous study showed that the RBS1 domain (part of Gemin5C) has a low to mid nanomolar range affinity to FMDV IRES. In this study, the affinity to the same RNA is around 50 μ M. This is very difficult to understand unless the

formation of Gemin5C decamer actually affects the affinity to RNA. This difference also brings back the question of whether a high-order decameric structure is naturally existing structure or not and the EMSA experiments were done properly.

Response: There are major differences in the assays that might contribute to explain the different affinities observed in earlier works: 1), the protein is different, 2), the quantification of the protein and the RNA probe also differs as they have been obtained by different methodologies. In all cases, the values reported are estimated from the experiments performed in each study.

For the G5C decamer, the structure of G5C decamer was solved by cryo-EM. Before frozen, the sample is a homogenous decamer, as evidenced by size-exclusion chromatography (SEC) and static light scattering (SLS) experiments.

- Specificity of the binding: Based on the argument made by the authors, Gemin5N specifically binds to the snRNP code of snRNAs. Thus, ideally, Gemin5N should be used as a negative control for at least first set of EMSA experiments.

Response: G5N has to be expressed and purified in large amount in insect cells, which takes another 2-3 months to get enough proteins. Previously, three independent studies already show that binding of Gemin5 WD40 domain to snRNA highly depends on the U-rich Sm site, and the binding of snRNA to Gemin5N was abolished by single nucleotide substitution (PMID: 27881600; 27881601; 27834343). Although we could not exclude the possibility that G5N can bind RNAs other than snRNAs, the RNA binding of G5N is highly selective. However, besides the U-rich sequence G5N specifically recognizes m7G cap of snRNAs (PMID: 27881600; 27881601; 27834343). Noticeably, during the foot-and-mouth disease virus (FMDV) infection, Gemin5 undergoes proteolysis to generate a transient C-terminal fragment spanning residues 845-1508 (PMID: 22362733) which interacts with the IRES element enhancing translation, suggesting that G5C serves as an independent functional module. Therefore, we believed that G5N is not required as a negative control in this manuscript.

To indicate specificity of the RNA binding of G5C, we measured the K_D between G5C and SL1, which is 6.5 μ M (new Fig. 1g). Also we found that a truncated form of SL1 bound to G5C 10-fold weaker. As the negative control, U₇ ssRNA does not bind to G5C (Fig. 1g). We believed that above new binding data supports RNA binding specificity of G5C.

3. The luciferase assay results in Figure 5 should be presented as SD not SEM. SEM tends to minimize the error bar and thus can make insignificant data to be statistically significant. Please also put the actually p-value instead of stating $p < 0.05$.

Response: This was done as requested (see revised Fig. 5a-b, Supplementary Fig. 7, and Supplementary Fig. 8a-b). However, for simplicity the actual p-values are given in the figure legend. Including the p-values on the figure resulted in an overcrowded image.

4. *The experimental design for luciferase assay lacks many controls. First, the authors should show how they controlled the transfection efficiency of both Gemin5 and its mutants expression and the luciferase plasmid. The current format only provides the expression level of Gemin5 and its mutants by western, which is pretty much eyeballing but not quantitative at all. The authors also should show western blot images for each biological repeat. More importantly, it is not clear how the authors are sure about the same transfection efficiency of luciferase plasmid in each Gemin5 and its mutants transfection. Another control missing in the experiment is the luciferase activity without Gemin5C binding domain. The fold changes in this type of experiments are misleading as they tend to hide a magnitude of data point fluctuation by the nature of sensitive assay. It would be better to show the normalized luciferase activity in the arbitrary unit (AU).*

Response: Quantitative data for the expression of the proteins in three independent assays is shown in the revised figure (**Fig. 5c-d**). The western blot images of all the experiments were used to obtain the quantitative data. The individual images are shown in **Supplementary Fig. 7**.

We have included experimental data (RT-qPCR) showing that the amount of mRNA expressed in transfected cells is similar for all the constructs (**Supplementary Fig. 8a-b**). Although there are small variations, no statistical differences were observed among the different constructs. The text has been modified in Results, and Materials and Methods.

As requested, the luciferase activity in Fig. 5a-b are plotted as arbitrary units (AU). The main point of this manuscript was to compare the activity of the mutants relative to the WT protein (**Fig. 5**). The vector alone was used as a control of the G5C expression, as shown in the WB images (**Fig. 5c-d, Supplementary Fig. 7**).

5. *The results shown in Figure 6 are intriguing, but they need more thoughtful interpretations. Why did the mutations in the TPR residues of the pocket affect the RNA binding while the mutations of RBS1 residues in the same pocket have no impact on RNA binding? Particularly, multiple studies reported RBS1 as an RNA-binding module in Gemin5C. Thus, the authors should explain the discrepancy between their findings and others.*

Response: Thanks for this comment, which indicates that we may have not been able to express ourselves correctly. In previous studies, the protein carrying only the RBS1 domain was shown to interact with RNA (**PMID: 32476560**) while the protein containing only the TPR module did not (**PMID: 31799608**). In Fig. 6 we are showing that some residues of the RBS1 domain (encompassing the unstructured region between α helices 26 and 27, which includes the disordered region previously reported to be involved in RNA binding (**PMID: 34424823**) are located near positively charged residues of the TPR module, forming a channel large enough to allow the assembly of a complex with SL1 RNA. Consistently, the cryo-EM structure of G5C decamer also reveals that the RBS1-containing loop is intrinsically unstructured (**Fig. 6c**), which might be resolved upon RNA binding. In addition, we provide evidence for the requirement of these positively charged residues to allow RNA binding (**Fig. 6e**), while

the overall decamer conformation is maintained (**Fig. 6d**). To reinforce these results, we have added a new mutant G5C-A951E, showing that impairing dimerization also reduces RNA binding (**new Fig. 6 f-g**).

In sum, the TPR dimer alone does not bind to RNA, however, in the highly assembled G5C decamer structure, TPR domain from one dimer (B-B') coordinates with the RBS1 from adjacent dimer (C-C') to bind RNA ligands. We have modified this paragraph accordingly.

6. In their model shown in Figure 7c, the authors used PTB as an example. Ironically, this model does not consider the affinity of RNA binding proteins to their target RNA at all. PTB is known to be specific to its target sequences with a high affinity. It is almost impossible for mid to high μM affinity interaction (Gemin5C to target RNA binding) can compete with such a high affinity interaction made by PTB. Either the authors should revise their model or should provide evidence that Gemin5C indeed can outcompete the PTB protein for their RNA binding.

Response: The reviewer raised a valid point, which indicates that we may have not been able to express ourselves correctly. In previous studies, the competition between PTB and G5₁₂₈₇₋₁₅₀₈ for RNA binding was performed by SHAPE footprint (**PMID: 23221641**). Comparison of the binding affinity of PTB and G5 for a given RNA using different methodologies and distinct protein regions is unacceptable. Therefore, we removed panel C from Figure 7. Also, while we show here that G5CT forms a decamer, how G5NT is placed in the overall structure remains elusive. Figure 7A depicts a simplified cartoon of a proposed tentative model that awaits future work.

Minor points)

- 1. There are incomplete sentences in the manuscript.*
- 2. In supplementary Figure 5, the second sequence alignment from the bottom has a wrong labeling on Gallus gallus. Other alignments used Gg but the second one from the bottom alignment used Ch (Chicken).*

Response: Thanks for these remarks. They were corrected in the revised manuscript.

Reviewer #3 (Remarks to the Author):

In this interesting manuscript, Guo et al. report the three-dimensional structure of the C-terminal region of Gemin5 (G5C). Gemin5 is an essential component of the SMN complex, an RNA-binding protein involved in the formation of small nuclear ribonucleoproteins. It is shown that G5C adopts a homodecameric structure assembled from a dimer of pentamers. The structure was thoroughly validated using mutagenesis together with in vitro and in cell assays. It is convincingly shown that a large positively charged concave surface brought about by G5C oligomerization is needed for the interaction of Gemin5 with stem-loop-containing RNA molecules. Based on the G5C structure, the authors make a good case that some of the previously identified pathogenic mutations in Gemin5 probably prevent pentamer formation and RNA binding. In conclusion, Guo et al. have uncovered a unique oligomeric arrangement of

G5C that is key for binding stem-loop regions of RNAs. The new structural knowledge resulting from this carefully done study provides a rigorous framework for testing hypotheses related to the functions and modes of action of Gemin5 in SMN. This work is of high technical quality and is appropriate for publication in Nature Communications.

Response: We greatly appreciate your positive feedback on our current work, and agree that our work would be of great interest to a broad audience. We have fully addressed the comments raised by the reviewer.

Minor points:

SL1 (stem-loop 1) should be defined in the text.

Response: Thanks for raising this point. SL1 is now defined as nt 3959-4044 of Gemin5 mRNA (NM_015465.5) in the revised manuscript. Accordingly, SL1 were numbered as in Gemin5 mRNA in revised Fig. 1b.

In Supplementary Table 1, an FSC threshold of 0.125 is indicated for the Gemin5 (841-1508) protomer. Where does this value come from? Is it a typo?

Response: It was a typo and was corrected in Supplementary Table 1.

Reviewer #4 (Remarks to the Author):

The authors determined the structure of Gemin5 C-terminal domain that behaves as a decamer. Previously, it has been known that Gemin5 N-terminal WD40 recognizes snRNAs to ensure the assembly of snRNPs. This study dissects the mRNA binding specificity of Gemin5 CT. They uncovered the molecular mechanism underlying decamer formation and examined the potential mRNA binding interface on Gemin5 CT. Overall, the cryo-EM structure is of high quality and these data must be valuable for the researchers in the RNA field. However, some issues should be addressed before it could be accepted for publication.

Response: We are grateful the reviewer for going through our paper in-depth and giving his/her constructive feedback that allowed us to improve the quality and presentation of work.

1. Is that structure similar to any solved structures? Can they compare the Gemin5 decamer with other similar modules?

Response: By using DALI server, we did not identify any homology structure that has > 30% identity to G5C. However, the homodecamer architecture was those found in NLRP3 and cyanase. We added a figure to compare those homodecamer structures (**Supplementary Figure 5**).

2. The authors examined the binding between Gemin5 and SL1, and Gemin5 and IRES. The sequences of the two RNAs are quite different. Could the authors further generate some RNA variants to examine whether the binding is based on the RNA structure, such as the length of stem loop?

Response: Our previous work indicated that the RBS1 domain does not interact with short ssRNA, such as UUUCCUUU under the same conditions used for SL1 binding (**PMID: 32476560**). In addition, SL1 apparently binds to G5C stronger than D5-IRES and SL1 does contain larger stem loop regions (**revised Fig. 1f**), indicating that G5C prefers RNA ligands with longer stem loops. Consistently, a truncated form of SL1 (SL1^{short}) bound to G5C 10-fold weaker than SL1 (K_{DS} : 67 μ M vs. 6.5 μ M) (**new Fig. 1g**). Further study on G5C-RNA complex is required to fully unravel its RNA binding property.

3. There are some loops invisible in the cryo-EM structure of Gemin5, it is not clear whether they also play roles in RNA binding.

Response: We thank the reviewer for pointing it out. As shown in Fig. 6, two loops are invisible in our structure. One is between $\alpha 19$ and $\alpha 20$, the other is between $\alpha 26$ and $\alpha 27$. Our previous work found that RBS1 resides in the loop between $\alpha 26$ and $\alpha 27$, and plays an important role in binding to cognate RNA ligands (**PMID: 29771365**). The loop between $\alpha 19$ and $\alpha 20$ is close to RBS1 and it might also contribute to RNA binding. We discussed the potential role of the loops in the revised manuscript.

4. For the cryo-EM map reconstitution and refinement, there should be an FSCwork/FSCfree curve to show the absence of overfitting in model refinement. A model vs map curve should also be reported.

Response: As the reviewer suggested, we have added the model vs map (work and free) FSC curves (see **revised Supplementary Fig. S2g**).

REVIEWERS' COMMENTS

Reviewer #2 (Remarks to the Author):

The authors' responses to my critiques were satisfactory. The manuscript improved its quality significantly. I support the manuscript to be published.

Reviewer #4 (Remarks to the Author):

The revised version of manuscript has addressed my concerns. I suggest to accept the interesting paper and publish it as soon as possible.

REVIEWERS' COMMENTS

Reviewer #2 (Remarks to the Author):

The authors' responses to my critiques were satisfactory. The manuscript improved its quality significantly. I support the manuscript to be published.

Reviewer #4 (Remarks to the Author):

The revised version of manuscript has addressed my concerns. I suggest to accept the interesting paper and publish it as soon as possible.

Response: We thank the reviewers for their positive comments on our revised manuscript.